# Omics Analysis Unveils the Pathway Involved in the Anthocyanin Biosynthesis in Tomato Seedling and Fruits

**DOI:** 10.3390/ijms24108690

**Published:** 2023-05-12

**Authors:** Rui He, Kaizhe Liu, Shuchang Zhang, Jun Ju, Youzhi Hu, Yamin Li, Xiaojuan Liu, Houcheng Liu

**Affiliations:** College of Horticulture, South China Agricultural University, Guangzhou 510642, China; ruihe@stu.scau.edu.cn (R.H.); 1836945107@stu.scau.edu.cn (K.L.); shuchangzhang@stu.scau.edu.cn (S.Z.); jujun-mail@stu.scau.edu.cn (J.J.); youzhihu@stu.scau.edu.cn (Y.H.); yaminli@stu.scau.edu.cn (Y.L.); liuxjjy628@stu.scau.edu.cn (X.L.)

**Keywords:** anthocyanin biosynthesis, purple tomato, HY5, MBW, PIFs

## Abstract

The purple tomato variety ‘Indigo Rose’ (InR) is favored due to its bright appearance, abundant anthocyanins and outstanding antioxidant capacity. *SlHY5* is associated with anthocyanin biosynthesis in ‘Indigo Rose’ plants. However, residual anthocyanins still present in *Slhy5* seedlings and fruit peel indicated there was an anthocyanin induction pathway that is independent of HY5 in plants. The molecular mechanism of anthocyanins formation in ‘Indigo Rose’ and *Slhy5* mutants is unclear. In this study, we performed omics analysis to clarify the regulatory network underlying anthocyanin biosynthesis in seedling and fruit peel of ‘Indigo Rose’ and *Slhy5* mutant. Results showed that the total amount of anthocyanins in both seedling and fruit of InR was significantly higher than those in the *Slhy5* mutant, and most genes associated with anthocyanin biosynthesis exhibited higher expression levels in InR, suggesting that *SlHY5* play pivotal roles in flavonoid biosynthesis both in tomato seedlings and fruit. Yeast two-hybrid (Y2H) results revealed that *SlBBX24* physically interacts with *SlAN2-like* and *SlAN2*, while *SlWRKY44* could interact with *SlAN11* protein. Unexpectedly, both *SlPIF1* and *SlPIF3* were found to interact with *SlBBX24*, *SlAN1* and *SlJAF13* by yeast two-hybrid assay. Suppression of *SlBBX24* by virus-induced gene silencing (VIGS) retarded the purple coloration of the fruit peel, indicating an important role of *SlBBX24* in the regulation of anthocyanin accumulation. These results deepen the understanding of purple color formation in tomato seedlings and fruits in an *HY5*-dependent or independent manner via excavating the genes involved in anthocyanin biosynthesis based on omics analysis.

## 1. Introduction

Anthocyanins comprise a class of primary hydrosoluble pigments belonging to flavonoids, which are widely distributed in plants and confer various colorations in fruit, flower, seed and leaf. Anthocyanins are essential antioxidants that not only play their crucial roles in protecting plants from various biotic and abiotic stressors (i.e., cold, drought, UV irradiation, pathogen) but also contribute to decreasing the risk of certain types of cardiovascular and neurodegenerative diseases and cancer in the human body [1,2,3]. Anthocyanin biosynthetic pathways have been intensively studied in many species, and various structural genes and transcription factors have been well characterized in a strongly conserved pathway [4,5,6]. The regulation of anthocyanin biosynthesis is controlled by early biosynthetic genes (i.e., phenylalanine ammonia-lyase (*PAL*), 4-coumaryl:CoA ligase (*4CL*), chalcone synthase (*CHS*), chalcone isomerase (*CHI*) and flavanone 3-hydroxylase (*F3H*)) and late biosynthetic genes (i.e., flavonoid 3′5′-hydroxylase (*F3′5′H*), dihydroflavonol 4-reductase (*DFR*), anthocyanidin synthase (*ANS*), glutathione-S-transferase (*GST*) and flavonol-3-glucosyltransferase (*3GT*)) [7]. Most genes involved in anthocyanins biosynthesis could be activated or repressed by specific transcription factors as well as controlled by a ternary MYB–bHLH–WD repeat (MBW) transcriptional complex, which consists of basic helix-loop-helix (bHLH) R2R3-MYB transcription factors and WD40-repeat proteins [8]. In addition, other regulatory factors, such as HY5, ERFs, PIFs, BBXs, and WRKYs, also participated in the regulation of anthocyanin biosynthesis [9,10,11,12]. Previous studies have proved that *AtBBX21*, *AtBBX22* and *AtBBX23* induced the accumulation of anthocyanins in Arabidopsis [13,14,15,16], while *AtBBX24*, *AtBBX25* and *AtBBX32* inhibit anthocyanin accumulation [17,18,19]. *SlBBX20* directly binds the promoter of the anthocyanin biosynthesis gene *SlDFR* to enhance anthocyanin biosynthesis in tomato fruits [20]. *MdBBX22* induced *mdm-miR858* expression via bounding to its promoter, thus governing anthocyanin accumulation in apples [21]. Additionally, members of the B-box (BBX) protein family (i.e., *BBX18/20/21/23/24/33*) directly conjunct with HY5 cooperatively regulate anthocyanin synthesis in Arabidopsis [22]. Furthermore, results showed that a WRKY gene negatively regulates the complex MYB-bHLH-WD40 petunia (*Petunia hybrida*) and Arabidopsis thaliana [23]. *PpWRKY44* could significantly promote light-induced anthocyanin accumulation in red pear fruit via binding to the promoters of *PpMYB10* [24].

The tomato (*Solanum lycopersicum*) is one of the most consumed vegetable products around the world. In most of the cultivated tomatoes, anthocyanins are generally undetectable in fruit. Cultivation attempts have been made to improve the anthocyanin content in tomato fruit. ‘Indigo Rose’ (InR), a purple tomato variety that contains the Aft locus and recessive atv locus, exhibits a high-level accumulation of anthocyanin on the fruit peel in a light-dependent manner [25]. Thus, ‘Indigo Rose’ has been frequently taken to underly the molecular mechanism of anthocyanin synthesis in purple tomato fruit [26,27,28]. Previous studies revealed R2R3-MYB transcription factor *SlAN2-like* as an active and critical regulator of anthocyanin biosynthesis, while *SlMYBATV* was identified as the regulatory repressor via competing for the binding of *SlAN2-like* to *SlAN1* [27].

HY5 (Elongated Hypocotyl 5), as a vital regulator of light-dependent development in higher plants, exhibits a dominant function in hypocotyl elongation and lateral root development as well as pigment accumulation [29]. To date, most of the genes and transcription factors involved in anthocyanin regulation were highly associated with HY5. The HY5 protein directly binds to either G-box or ACE-box in the promoters of anthocyanin biosynthetic genes such as *CHS* and *DFR,* then activates their expression, positively regulating anthocyanin accumulation [30]. *CaHY5* can bind to the promoter of *CaF3H*, *CaF3′5′H*, *CaDFR*, *CaANS* and *CaGST*, which are well related to anthocyanin biosynthesis or transport, and thereby promote anthocyanin accumulation in pepper hypocotyl [31]. However, residual anthocyanins have still been present in *hy5* mutants of Arabidopsis, which indicates that there is an anthocyanin induction pathway that is independent of HY5 in plants [30,32]. This result has been well proved in *Slhy5* mutants of tomatoes via analyzing the transcriptome of multiple tissues, which has found eight candidate transcription factors were likely involved in anthocyanin production in tomatoes in an HY5-independent manner [11]. In the present study, we found that anthocyanins accumulated on the surface of hypocotyls in InR tomato seedlings, but not *Slhy5* seedlings, at cotyledon emergence. Unexpectedly, residual anthocyanins also accumulated both in the cotyledon and hypocotyls of *Slhy5* seedling, which displayed obvious spatiotemporal specificity. Meanwhile, the InR fruit peel accumulated large amounts of anthocyanins, particularly in the light-exposing part, while *Slhy5* contained a lower anthocyanin content on the peel of the fruit shoulder and no anthocyanins accumulated in the peel of the shading part. Therefore, whether other transcription factors substitute or compensate for HY5 to regulate the anthocyanin accumulation? Advances in transcriptomics and metabolomics play pivotal roles in uncovering complex biological mechanisms of diversified pathways in plants [33,34].

The objectives of this study were to reveal the anthocyanin variation in seedlings and fruit of tomato (‘Indigo Rose’ (InR) and *Slhy5* mutants) at different developmental stages and excavate the candidate *HY5*-dependent or independent transcription factors involved in anthocyanin biosynthesis in the seedling and fruit via omics analysis.

## 2. Results

### 2.1. Morphological Characterization of InR and Slhy5 Seedling and Fruit

On the stage in which hypocotyl emergence and cotyledons still close, the surface of hypocotyls of InR seedlings exhibited purple color, whereas *Slhy5* seedlings displayed white color (hardly accumulate anthocyanins) and longer hypocotyl (Figure 1a and Appendix A). Considerable anthocyanins accumulation was observed in cotyledons and hypocotyls of InR seedlings once exposed to light (Figure 1a and Appendix A). However, *Slhy5* seedlings developed an opposite phenotype and displayed obvious spatiotemporal specificity in anthocyanin production. Briefly, in *Slhy5* seedlings, the anthocyanin accumulated first in the upper part of the hypocotyls and then gradually developed in the lower part in a light-dependent manner (Figure 1a and Appendix A). The anthocyanin content both in cotyledons and hypocotyls of *Slhy5* seedlings was lower than those of InR seedlings, respectively (Figure 1c and Appendix A).

Anthocyanin content just accumulated in peels at the green-mature stage and fully mature stage in InR and *Slhy5* fruits, and few anthocyanins were found in the fruit flesh (Figure 1b,d and Appendix A). Additionally, higher anthocyanins accumulated in the light-exposing peel part of InR fruits than in the shading part (Figure 1b,d). Unexpectedly, residual anthocyanins have still been present in the peel of the *Slhy5* fruit shoulder, though the anthocyanins contents were lower than the InR fruit peel. However, anthocyanin accumulation was hardly detected in the shading peel part of the *Slhy5* fruit (Figure 1b,d). These observations suggested that *Slhy5* had a predominant role in tomato pigmentation in a light-dependent manner, and there might be some regulators controlling anthocyanin biosynthesis in an HY5-independent manner.

### 2.2. Changes of Metabolites and Genes Expression in the Cotyledon of InR Seedlings and Slhy5 Seedlings

The color of cotyledon and hypocotyl was transformed continuously from green to purple during seedling development in InR seedlings and *Slhy5* seedlings. To explore the changes of metabolites and gene expression during InR and *Slhy5* seedling development, metabolic and transcriptome analysis of cotyledon, the upper and lower part of the hypocotyl were carried out at different developmental stages, respectively (Figure 1a). PCA was performed on detected metabolites to demonstrate the similarity in metabolic profiles among the samples (Appendix A). In the two-dimensional PCA plot, three biological replicates of each sample tended to group, indicating the high reproducibility of the generated metabolome data (Appendix A). Many metabolites and genes in the seedling varied considerably in terms of different tissues of different varieties. Therefore, variation of metabolites and gene expression in three parts (cotyledon, upper and lower part of the hypocotyl) of InR seedlings and *Slhy5* seedlings was investigated, respectively.

Four metabolite groups were observed in cotyledon based on the level of annotation of metabolite similarity. Group 1, which metabolites levels of InR seedlings cotyledon (W_2_C) were obviously higher than those in *Slhy5* seedlings (H_2_C, H_3_C and H_4_C), including tulipanin, rutin, butin etc., belonged to flavonoids, flavones and flavonols, anthocyanins according to KEGG analyses (Appendix A). So, anthocyanins and flavonoids might be responsible for the distinction of purple coloration of cotyledon between InR and *Slhy5* seedlings.

The transcriptome data validated the authenticity and accuracy of the metabolic analysis. A weighted gene co-expression network analysis (WGCNA) was performed on the genes of cotyledon. A slight relationship with purple coloration was displayed in the green module (Figure 2b), and the genes related to this module were annotated according to KEGG pathway enrichment analysis, which involved the flavonoid biosynthesis pathways (Figure 2c). Anthocyanins metabolism-related transcriptional factors (i.e., *SlAN1*, *SlAN2*, *SlAN2-like*) and the structural genes (i.e., *SlPAL*, *Sl4CL*, *SlCHS*, *SlCHI*, *SlDFR*, *SlANS*) displayed higher expression levels in the cotyledon of InR seedlings than *Slhy5* seedlings (Figure 2c,d). Meanwhile, the expression of these genes increased with the development of *Slhy5* seedlings cotyledon, which is consistent with the increasing trend of flavonoid and anthocyanins contents (Figure 2c,d).

### 2.3. Changes of Metabolites and Genes Expression in the Upper Part of the Hypocotyl of InR Seedlings and Slhy5 Seedlings

In the upper part of hypocotyl, the differential metabolites in group 1, which was positively correlated and had similar consistent patterns with the purple coloration in the hypocotyl, also contained a variety of flavonoids such as petunidin 3-O-glucoside, tulipanin, isotrifoliin (Figure 3a). The KEGG enrichment analysis displayed that the terms ‘Flavone and flavonol biosynthesis’, ‘Photosynthesis’, ‘Anthocyanin biosynthesis’, ‘Vitamin B6 metabolism’, and ‘Starch and sucrose metabolism’ were significantly enriched in the Group 1 (Appendix A). WGCNA identified genes in the black module with significant co-expression with the biosynthesis of the metabolite in the flavonoid pathway, which was responsible for the purple coloration observed in the upper part of the hypocotyl (Figure 3b).

KEGG enrichment analysis was performed on the genes in the black module and showed that these DEGs were enriched mainly in flavonoid biosynthesis pathways (Figure 3c). Consistent with the data from the transcriptomic analysis, both the anthocyanin positive regulatory genes, such as *SlAN2*, *SlAN2-like*, *SlAN1I* and *SlAN11* (except for the negative regulatory genes *SlMYB7*, *SlMYB3* and *SlMYB32*) and the anthocyanin biosynthetic genes, including *SlPAL*, *Sl4CL*, *SlCHS*, *SlCHI*, *SlF3H*, *SlF3’H*, *SlDFR*, *SlANS* and *Sl3GT* (Figure 3d,e), exhibited much higher expression in the upper part of InR seedlings hypocotyl than those of *Slhy5* seedlings. Compared with H_1_HU and H_2_HU, the genes involved in anthocyanins under H_4_HU and H_3_HU displayed higher expression levels (Figure 3d,e).

### 2.4. Changes of Metabolites and Genes Expression in the Lower Part of the Hypocotyl of InR Seedlings and Slhy5 Seedlings

Similarly, annotated metabolites in the lower part of the hypocotyl could be divided into several large groups based on similar variation tendencies, respectively. Among these metabolites, group 1 metabolites were present in higher levels in InR seedlings than in *Slhy5* seedlings, which were consistent with the color variations during seedling development. These metabolites include petunidin 3-O-glucoside, tulipanin and isotrifoliin (Figure 4a). KEGG analysis showed that these metabolites in different parts of the seedling were mainly enriched in flavone and flavonol biosynthesis and anthocyanin biosynthesis (Appendix A). The genes in mode marked by pale turquoise color according to WGCNA analysis of transcriptomic data from the lower part of the hypocotyl were consistent with the increasing trend of anthocyanin contents (Figure 4b). The KEGG enrichment analysis showed that the term ‘flavonoid biosynthesis’ pathway was significantly more pronounced (Figure 4c). Meanwhile, the expression levels of anthocyanin biosynthesis genes in the lower part of InR hypocotyl were higher than *Slhy5* (Figure 4d,e). In addition, the expression of most anthocyanin biosynthetic genes in the lower part of the *Slhy5* seedling hypocotyl under different stages exhibited the following trend: H_4_HD ≈ H_3_HD > H_2_HD > H_1_HD.

### 2.5. Changes of Metabolites and Genes Expression in Different Parts of Slhy5 Seedling

Based on WGCNA analysis, we identified a module (marked in cyan) whose gene expression pattern was associated with the phenotype of anthocyanin synthesis in *Slhy5* seedlings at the third and fourth development stages (Figure 5a,b). Twenty-one co-expressed genes in the cyan module were significantly correlated with pigment accumulation in *Slhy5* seedlings. These indicated that bHLH (*SlAN1*) was a hub gene involved in the positive regulation of flavonoid metabolism in *Slhy5* seedlings (Figure 5c), possibly by affecting structural node genes, such as *SlCHI*, *SlCHS*, *SlAN3*, *SlDFR* and *SlRT*, etc.

### 2.6. Screening of Differentially Expressed Genes of Tomato Fruit

Similar to seedlings, residual anthocyanin production was also observed in the peel of the *Slhy5* fruit shoulder, whereas anthocyanin was almost undetectable in the shading part of the *Slhy5* fruit (Figure 1b,d). To underly gene expression changes over the fruit peel of InR and *Slhy5*, RNA-Seq analysis was conducted. The number of differentially expressed genes had a very high variance among InR and *Slhy5* fruit peel. Regarding WT-S-vs-slhy5-S (purple-colored peel of InR fruit compared to purple-colored peel of *Slhy5* fruit), a total of 2995 DEGs, including 1360 up- and 1635 down-regulated genes were detected (Appendix A). These genes were enriched in KEGG pathways related to photosynthesis, carbon fixation in photosynthetic organisms, phenylpropanoid biosynthesis, phenylalanine metabolism, flavonoid biosynthesis and flavone and flavonol biosynthesis (Figure 6a), which the gene FKPM values in InR fruit peel was higher than those of *Slhy5* (Figure 6c,d). These results well demonstrated the importance of HY5 in the color formation (both green and purple) of tomato fruits.

Meanwhile, to further investigate the DEGs related to *Slhy5* fruit peel coloration, we compared the FPKM values of Slhy5-S-vs-Slhy5-N (purple-colored peel compared to white-colored peel in *Slhy5* fruit). A total of 2393 DEGs were identified from the groups of Slhy5-S-vs-Slhy5-N (Appendix A). According to KEGG enrichment analysis, the 20 top-ranked pathways contributed by these DEGs were photosynthesis, carbon fixation in photosynthetic organisms, porphyrin and chlorophyll metabolism, flavonoid biosynthesis, phenylpropanoid biosynthesis, flavone and flavonol biosynthesis, carotenoid biosynthesis and phenylalanine metabolism. A set of genes involved in flavonoid metabolism in the light-exposing part of the *Slhy5* fruit peel displayed a higher expression level than the shading part (Figure 6b). Therefore, we predicted that the flavonoid biosynthesis pathway was leading to the purple peel coloration in *Slhy5* fruit peel, that was proved that other genes or TFs were involved in anthocyanin biosynthesis in an HY5-independent manner.

To further verify expression patterns of the DEGs in seedlings and fruit, the genes *SlAN1*, *SlPAL*, *SlCHS*, *SlCHI*, *SlF3H*, *SlDFR*, *SlANS*, *Sl3-GT*, *SlAAC* and *SlGST* were used for qRT-PCR verification (Appendix A). The qRT-PCR results were consistent with the transcriptomic analysis results.

### 2.7. The Genes Involved in the MYB-bHLH-WD40 (MBW) Complex That Activates Anthocyanidins in Tomato Fruit

To explore the regulation of flavonoid metabolism, the possible interaction of flavonoid metabolism-related genes was tested utilizing yeast two-hybrid (Y2H) assays. *SlPIF1*, *SlPIF3*, *SlAN2-like*, *SlAN2*, *SlAN11*, *SlAN1*, *SlHY5*, *SlJAF13*, *SlBBX24* and *SlWRKY44* were selected for the candidate genes based on the RNA Sequencing results, which might contribute to anthocyanin biosynthesis. We found that SlBBX24 physically interacts with SlAN2-Like and SlAN2 but not with SlAN11 in yeast, while SlWRKY44 could interact with SlAN11 protein only and showed no affinity for other genes. Additionally, these results indicated that SlBBX24 could interact with SlPIF1 and SlPIF3 but not SlPIF4. Unexpectedly, SlAN1 and SlJAF13 displayed the same physical interaction, which could interact with both SlPIF1 and SlPIF3 (Figure 7).

### 2.8. SlBBX24 Physically Interacts with Regulators of Light Signaling and Anthocyanin Biosynthesis

To further explore whether SlBBX24 physically interacts with regulators of light signaling and anthocyanin biosynthesis, we conducted bimolecular luciferase complementation imaging (LCI) assays in Nicotiana benthamiana leaves. Luminescence was observed in leaves that co-expressed SlBBX20-nLUC and cLUC-SlHY5, cLUC-SlPIF1, cLUC-SlPIF3, cLUC-SlAN2-like, cLUC-SlAN2, cLUC-SlAN11. LUC signals were not detectable in the three controls. These results suggest that SlBBX24 associates with the transcription factors involved in the light-signaling pathway (SlHY5, SlPIF1, SlPIF3) and anthocyanin biosynthesis (SlAN2-like, SlAN2, SlAN11) in living plant cells (Figure 8).

### 2.9. SlBBX24 May Be Involved in Anthocyanin Accumulation in Tomato Fruit Peels

To investigate whether the biosynthesis of anthocyanin is regulated by candidate genes, a target gene of *SlBBX24* was silenced by the VIGS approach. Compared to the empty vector control, the expression of *SlBBX24* was reduced at 7 days after infiltration, and anthocyanin was less vibrant than that of control fruits (Figure 9). qRT-PCR analyses were performed to test the expression changes of the genes involved in anthocyanin biosynthesis. As shown in Figure 8, silencing of *SlBBX24* inhibited the expression levels of anthocyanin structural genes, including EBGs (*SlC4L and SlCHS1*) and LBGs (*SlDFR* and *SlANS*) as well as the positive regulators (*SlAN1*, *SlAN2*, *SlAN2-like*, *SlAN11*, *SlHY5*), and increased the expression levels of negative regulators (*SlMYBATV*, *SlTRY* and *SlMYB76*) (Figure 9). Therefore, it suggests that *SlBBX24* might be one of the important regulators in the regulatory chain of anthocyanin biosynthesis in tomato fruit.

**Figure 8 ijms-24-08690-f008:**
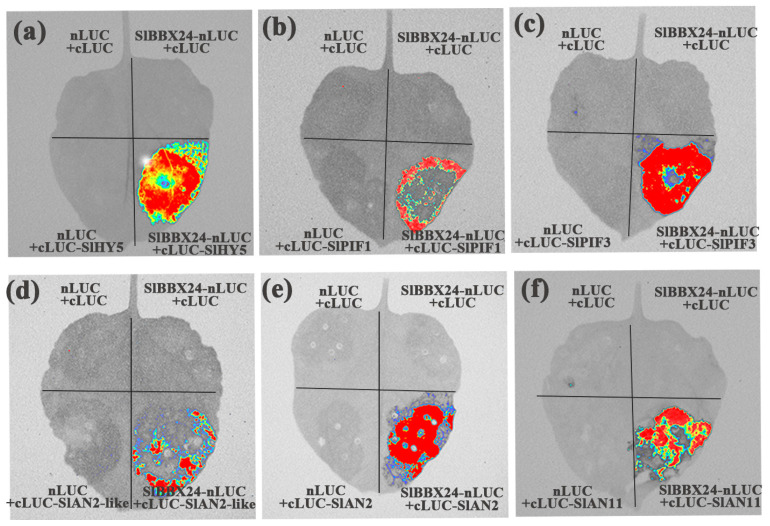
Interactions between *SlBBX20* and *SlHY5* (**a**), *SlPIF1* (**b**), *SlPIF3* (**c**), *SlAN2-like* (**d**), *SlAN2* (**e**), *SlAN11* (**f**) in firefly luciferase complementation imaging (LCI) assays in planta.

**Figure 9 ijms-24-08690-f009:**
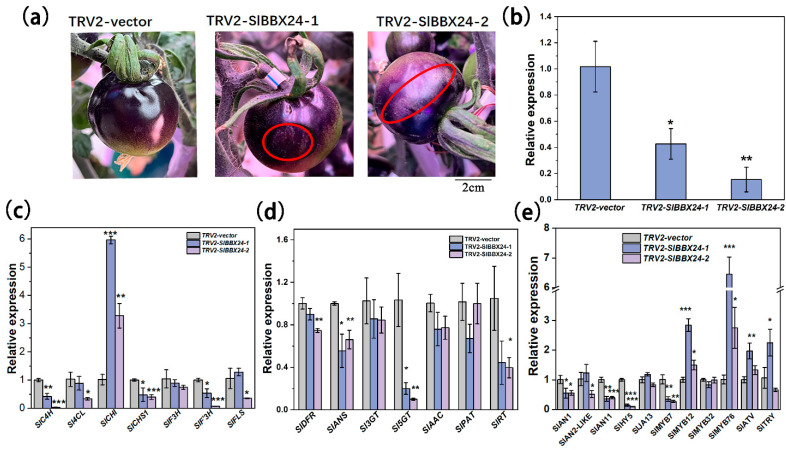
Analysis of VIGS of the *SlBBX24* gene. (**a**) Suppression of *SlBBX24* by virus-induced gene silencing (VIGS) retarded the purple coloration of the tomato fruit peel. (**b**) Relative expression of *SlBBX24* in fruit peels of TRV2-inoculated and TRV2-SlBBBX24-inoculated for 7 days. (**c**–**e**) Relative expression of anthocyanin structural genes of fruit peels of TRV2-inoculated and TRV2-*SlBBX24*-inoculated for 7 days. Statistically significant differences between the purple zone and the green zone were determined by Student’s *t*-test. (*** *p* < 0.001, ** *p* < 0.01, * *p* < 0.05).

## 3. Discussion

### 3.1. Flavonoids Might Be Attributed to Major Color Differences among InR and Slhy5 Mutant Seedlings and Fruit Peel

In recent years, integrated metabolomic and transcriptomic analyses as efficient tools to underly the molecular mechanisms of key metabolic pathways in plants [35,36,37,38]. A detailed description of secondary metabolic changes occurring in the whole germinated seeds as well as cotyledons, hypocotyls and roots from 3 to 9 days old tomato seedlings via LCMS profiling, which provided a new perspective to study metabolic networks controlling flavonoid biosynthesis in tomato [39]. The metabolite variants across 20 major tomato growth tissues and stages were explored by combining transcriptome and metabolome approaches, which verified novel transcription factors that control steroidal glycoalkaloids and flavonoid pathways [40]. In the present study, an integrated analysis of the transcriptome and metabolome was conducted to reveal the differences between wild-type (InR) and *Slhy5* seedlings at different growth and development stages. A total of 987 metabolic components were accumulated specifically in InR and *Slhy5* seedlings, of which amino acids and organic acids, flavanones, flavones and isoflavonoids accounted for a large proportion (Appendix A). Flavonoids, a product of the phenylpropanoid metabolism pathway, are extensively distributed in numerous plants and are composed of various subclasses, including flavanones, flavones, isoflavonoids, anthocyanins and flavonols. Anthocyanins, as a key flavonoid subgroup, are responsible for pigmentation in flowers, fruit, seed, and leaf [4]. Our metabolic profiling found metabolites in different parts of the seedling was mainly enriched in flavone and flavonol biosynthesis (Figure 2). Meanwhile, levels of flavonoids, including anthocyanins, in InR seedlings were obviously higher than those of *Slhy5* (Appendix A). A KEGG analysis in WT-S-vs-Slhy5-S, Slhy5-S-vs-Slhy5-N were related to phenylpropanoid biosynthesis, phenylalanine metabolism, flavonoid biosynthesis and flavone and flavonol biosynthesis (Figure 5a). These genes were enriched in KEGG pathways related to photosynthesis, carbon fixation in photosynthetic organisms, phenylpropanoid biosynthesis, phenylalanine metabolism, flavonoid biosynthesis and flavone and flavonol biosynthesis (Figure 5a), and the gene FKPM values in InR fruit peel was higher than those of *Slhy5*. These results were well indicated that the main pigment components in InR and *Slhy5* seedlings and fruit peel were flavonoids.

### 3.2. SlHY5 Acts as a Master Regulator to Control Anthocyanin Biosynthetic in Seedlings and Fruit of Tomato

Anthocyanin biosynthetic genes are regulated directly by the MBW complex consisting of MYB, bHLH and WDR proteins. Aft and Atv are two important loci that are well-associated with anthocyanin biosynthesis in tomatoes. MYB TFs occupy the major determinant position in the control network of anthocyanin biosynthesis and have been well demonstrated [41,42]. Four R2R3 MYB TF genes (*SlAN2*, *SlANT1*, *SlANT1-like* and *SlAN2-like*) were previously identified to regulate anthocyanin biosynthesis in tomatoes [43,44,45]. Two bHLH TFs, *SlAN1* and *SlJAF13*, were recently reported to regulate anthocyanin production [46]. Otherwise, a tomato WDR protein, *SlAN11,* was also involved in anthocyanin synthesis [47].

HY5 was well characterized as a positive regulator of anthocyanin synthesis in a light-dependent manner. Knock-down *SlHY5* transcription significantly reduced the anthocyanin levels both in seedlings and fruit of tomatoes [11]. *HY5 is* involved in the regulation of anthocyanin biosynthesis by directly binding to MYB transcription factors, such as the production of flavonol glycosides *(MYB12/PFG1* and *MYB111/PGF3*), production of anthocyanin pigment1 (*MYB75/PAP1*) and MYB-like Domain (*MYBD*) [48]. *HY5* activates the expression of *PAP1* expression via direct binding to G-and ACE-boxes in the promoter region, positively inducing the accumulation of anthocyanin in Arabidopsis [22]. Consistently, *PyHY5* alone or interacted with *PyBBX18* activates the expression of *PyMYB10* and *PyWD40,* which subsequently regulate the anthocyanins accumulation in red pear [49]. In this study, expression levels of *SlAN2-like*, *SlAN2*, *SlAN1* and *SlAN11* were higher in InR than *Slhy5* both in seedlings and fruit, and the expression pattern of these genes was consistent with pigment accumulation (Figure 2, Figure 3, Figure 4 and Figure 6). Two-hybrid (Y2H) assays determined *SlHY5* regulated anthocyanin biosynthesis through interaction with *SlAN2* (Figure 7), consistent with the result that *SlHY5* is a positive regulator of anthocyanin biosynthesis in vegetative tissues of tomato [50]. Therefore, *SlHY5* might be the master regulator to control anthocyanin accumulation in InR seedlings and fruit via mediating the transcriptional activity of an MBW complex and the enhanced expression of key genes, such as *SlCHI*, *SlCHS*, *SlF3H*, *SlDFR* and *SlANS*.

### 3.3. Possible Regulatory Mechanisms of Anthocyanin Biosynthesis in an HY5-Independent Manner in Tomato

Besides MYB and bHLH, other TF families also regulate anthocyanin biosynthesis. *SlBBX20* could directly bind to the promoter of *SlDFR* to activate its expression, thus promoting anthocyanin accumulation in tomatoes [20]. In apple, *MdBBX37* interacted with *MdMYB1* and *MdMYB9*, two key positive regulators of anthocyanin biosynthesis, and inhibited the binding of those two proteins to their target genes and, therefore, negatively regulated anthocyanin biosynthesis [10]. *PpBBX18* and *PpBBX21* antagonistically regulate anthocyanin biosynthesis via competitive association with *PpHY5* in the peel of pear fruit. Also, discoveries have emphasized the importance of WRKY protein in the control of the flavonoid pathway and its relationship to the MBW complex [9]. *PpWRKY11* was able to bind to W-box cis-elements in the promoters of *PpMYB10,* then regulated anthocyanin synthesis in pear flesh [23]. In the present study, different from InR seedlings, some TFs were detected in *Slhy5* seedlings by RNA-Seq, such as WRKYs, BBXs and NACs, which might compensate for the function of HY5 and contribute to the expression of related genes involved in anthocyanin synthesis. Y2H assays revealed that *SlBBX24* could interact with *SlAN2-like* and *SlAN2*, which likely had a positive function in the regulation of anthocyanin biosynthesis (Figure 7 and Figure 8). In addition, *SlWRKY44* also could interact with the *SlAN11* protein (Figure 7), which was consistent with the result that the WRKY factor physically interacted with the AN11 in yeast two-hybrid analysis [51]. Silencing of *SlBBX24* via virus-induced gene silencing (VIGS) led to the downregulation of the expression of structural genes and caused a decrease in anthocyanin accumulation (Figure 9). Additionally, PIFs play a role in the biosynthesis of plant pigments. *PIF3* could specifically bind to the G-box element of anthocyanin biosynthesis-related structural genes promoter to up-regulate anthocyanin accumulation in an HY5-dependent manner under far-red light [30]. Furthermore, *PIF4* negatively regulated anthocyanin accumulation by inhibiting *PAP1* transcription in Arabidopsis seedlings [52]. In this study, *SlPIF1* and *SlPIF3* could physically interact with *SlAN1* and *SlJAF13*, as well as *SlBBX24*. We speculated that *SlPIF1*, *SlPIF3*, *SlBBX24* and *SlWRKY44* might be involved in anthocyanin biosynthesis in a manner independent or dependent on *SlHY5*.

Taken together, given the PIFs and MBW were the key regulators of anthocyanin biosynthesis, we proposed a model to clearly illustrate this mechanism (Figure 10). In InR, *SlHY5* expression was induced by light and then activated the activity of the MBW complex to regulate the anthocyanin accumulation. While in *Slhy5* seedlings and fruit, PIFs or several other transcription factors might be involved in coordinating anthocyanin biosynthesis, such as BBXs and WRKYs. More thorough and rigorous molecular studies should be performed to explore the relationship of PIFs or other transcription factors which might be involved in anthocyanin biosynthesis.

## 4. Materials and Methods

### 4.1. Plant Materials and Growth Conditions

The experiment was carried out in an artificial light plant factory at South China Agricultural University. The tomato seeds of wild-type (cv. ‘Indigo Rose’ (InR)) and *Slhy5* mutant were generously provided by Dr. Qiu of the College of Horticulture of South China Agriculture University [11]. Seeds were sanitized in 0.5% sodium hypochlorite for 15 min, then rinsed with distilled water. After being soaked in distilled water for 5 h at 25 °C, seeds were sowed in sponge cubes (2 cm × 2 cm × 2 cm) with distilled water in the plant growth chamber in the dark at 25 °C for 3 days. Then, the seedlings were grown under 300 µmol m^−2^ s^−1^ white LEDs (Chenghui Equipment Co., Ltd., Guangzhou, China; 150 cm × 30 cm), 10/14 h light/dark photoperiod, 24 ± 2 °C, and 65–75% relative humidity. The seedlings grown for the 4th, 5th, 6th and 7th days were sampled and divided into three parts: cotyledon (except for the samples on the 4th day, which cotyledons still closed), upper 1 cm of the hypocotyl, and bottom 1 cm of the hypocotyl (Figure 1a). Three biological replicates were collected for analyses, with each replicate composed of 60 seedlings. Fruits from the *Slhy5* mutants and ‘Indigo Rose’ were sampled at the green-mature stage and fully mature stage with three biological replicates (each replicate consisted of six fruits from different plants). The fruit peel and flesh were carefully split with a scalpel blade. The above-mentioned samples were immediately frozen in liquid nitrogen and stored at −80 °C for further analysis.

### 4.2. Anthocyanin Assay

The anthocyanin content was performed as described in a previous study with some modifications [53]. Samples (100 mg) were extracted with 1 mL buffers of pH 1.0 (50 mmol KCl and 150 mmol HCl) and pH 4.5 (400 mmol sodium acetate and 240 mmol HCl), respectively, and incubated overnight at 25 °C. The absorbance of the extract liquor was determined at 510 nm using a UV-spectrophotometer (UV-1600, Shimadzu, Kyoto, Japan).

### 4.3. RNA Sequencing and Data Analyses

Total RNA was extracted from different parts of the seedling (Figure 1a) and the peel of fruit (Figure 1b) using the RNeasy Plant Mini kit (Qiagen, Hilden, Germany). Total amounts and integrity of RNA were assessed using the RNANano 6000 Assay Kit of the Bioanalyzer 2100 system (Agilent Technologies, Santa Clara, CA, USA). Three independent biological replicates were performed. The RNA-seq sequencing and assembly of seedling and fruit peel were performed by NovoGene Science and Technology Corporation (Beijing, China) and Genedenovo Biotechnology Co., Ltd. (Guangzhou, China), respectively. A total of 3 μg RNA was prepared for sequencing libraries using the NEBNext UltraTM RNA Library Prep Kit for Illumina (NEB, Beverly, MA, USA) according to the manufacturer’s instructions and sequences attributed to each sample by adding index codes. The library preparations were sequenced on an Illumina Hiseq 4000 platform to generate paired-end reads. The raw sequence reads were filtered by removing adaptor sequences and low-quality sequences, and raw sequences were changed into clean reads. Then the clean reads were then mapped to the tomato reference genome sequence (ITAG 4.0) (https://solgenomics.net/organism/Solanum_lycopersicum/genome/, accessed on 18 November 2021). Padj ≤ 0.05 and |log2(foldchange)| ≥ 1 were set as the threshold for significantly differential expression. WGCNA analyses were constructed in the BioMarker cloud platform (http://www.biocloud.net, accessed on 10 March 2022).

### 4.4. Metabolite Extraction

Samples of different parts of seedling preparation, extract analysis, metabolite identification and quantification were performed by the NovoGene Database of NovoGene Co., Ltd. (Beijing, China). Tissues (100 mg) were individually ground with liquid nitrogen, and the homogenate was re-suspended with pre-chilled 80% methanol and 0.1% formic acid by vortexing. The supernatant was injected into an LC-MS/MS system. UHPLC-MS/MS analyses were performed using a Vanquish UHPLC system (ThermoFisher, Dreieich, Germany) coupled with an Orbitrap Q ExactiveTM HF mass spectrometer (ThermoFisher). Structural analysis of metabolites was determined using standard metabolic operating procedures. MRM was used to conduct metabolite quantification. All metabolites identified were subject to partial least squares discriminant analysis. Principle component analysis (PCA) and Orthogonal Partial Least Squares Discriminate Analysis (OPLS-DA) were carried out to identify potential biomarker variables. For potential biomarker selection, variable importance in projection (VIP) ≥ 1 and fold change (FC) ≥ 2 or ≤0.5 were set for metabolites with significant differences.

### 4.5. Yeast Two-Hybrid Assay

The amplified full-length CDSs of *SlPIF1*, *SlPIF3*, *SlAN2-like*, *SlAN2*, *SlAN11*, *SlAN1*, *SlHY5*, *SlJAF13*, *SlBBX24* and *SlWRKY44* were amplified and inserted into pGADT7 and pGBKT7 vectors, respectively. Primers used for amplified and plasmid construction are listed in Appendix A. Different combinations of bait and prey vectors were co-transformed into Y2Hgold and then cultured on SD/-Leu-Trp (SD-LT) medium supplemented at 28 °C for 2–3 days. Next, 2.5 μL of aliquots were patched on SD/Ade/His/Leu/Trp plates with 5-bromo-4-chloro-3-indolyl-α-D-galactopyranoside (X-α-Gal) and incubated at 28 °C for 3 days.

### 4.6. Virus-Induced SlBBX24 Gene Silencing in Tomato

Specific coding regions *SlBBX24* fragment were selected for VIGS vector construction by the VIGS design tool (http://solgenomics.net/tools/vigs, accessed on 10 November 2022). A 300 bp fragment of the coding region of SlBBX24 was amplified using the forward primer (5′-gtgagtaaggttaccgaattc ATGAAGATACAGTGTGATGTG-3′) and the reverse primer (5′-cgtgagctcggtaccggatcc AGTGGCTAAGAAGCGTTGGTG-3′). The PCR product was ligated into the *pTRV2* vector to construct the *TRV2::SlBBX24* vector. The recombinant plasmid and *TRV1* vector were transferred into *Agrobacterium GV3101*. Resuspensions of *pTRV1* and *pTRV2* (as a negative control) or its derivative vectors were mixed at a 1:1 ratio and then infiltrated into a mature green-stage tomato. The injected fruits were grown in a growth chamber at 22 ± 2 °C, and 12 h of light/12 h of darkness, and relative humidity was controlled in the range of 70 ± 5%. After two weeks, fruits were harvested and stored at −80 °C for RNA and qRT-PCR analysis to assess the degree of silencing.

### 4.7. Luciferase Complementation Imaging Assays (LCI)

The full-length CDS of *SlBBX24* was amplified and cloned, and ligated into pCAMBIA1300-35S-cLUC to produce SlBBX24-nLUC. *SlHY5*, *SlPIF1*, *SlPIF3*, *SlAN2-like*, *SlAN2*, *SlAN11* were fused with *pCAMBIA1300-35S-cLUC* to generate the *cLUC-SlHY5*, *cLUC-SlPIF1*, *cLUC-SlPIF3*, *cLUC-SlAN2-like*, *cLUC-SlAN2* and *cLUC-SlAN11*. Agrobacterium strains GV3101 containing the above constructs were infiltrated into the leaves of 6-week-old N. *benthamiana* plants. The leaf images were captured after a 1 d incubation in darkness at 22 °C and an additional 1 d incubation under a 16-h light/8-h dark photoperiod.

### 4.8. RNA Extraction and Quantitative Reverse Transcription PCR

Total RNA was extracted from different parts of the seedling and the peel of fruit using RNAex Pro Reagent (Accurate Biotechnology Co., Ltd., Changsha, China), and its quality and quantity were evaluated using a Nanodrop ND-1000 spectrophotometer (Thermo Fisher Scientific). The first cDNA strand was synthesized using Evo M-MLV RT for PCR Kit (Accurate Biotechnology Co., Ltd., Changsha, China). The relative expression levels were determined by performing a quantitative reverse transcription PCR (qRT-PCR) analysis using a LightCycler 480 system (Roche, Hercules, Switzerland) with an Evo M-MLV RT-PCR kit (Accurate Biotechnology Co., Ltd., Changsha, China). The PCR thermocycling protocol was as follows: 95 °C for 30 s, followed by 40 cycles of 95 °C for 15 s and annealing for 60 °C for 30 s. Three biological replicates and three qRT-PCR technical replicates were performed for each sample. The relative expression levels were normalized with the results of mean values of *SlUBI* using the 2^−ΔΔCt^ method [54]. The primer sequences used for the qRT-PCR analyses are listed in Appendix A.

### 4.9. Statistical Analysis

All values are shown as the means of three replicates with standard error (SE). Analysis of variance was performed by Duncan’s multiple range test using the SPSS 22.0 program (SPSS 22.0, SPSS Inc., Chicago, IL, USA). Different significance of means was tested by LSD at *p* < 0.05. Graphs were plotted using Origin 2021 (Origin Lab Corporation, Northampton, MA, USA). The heat map analysis was visualized by TBtools software (TBtools-II v1.120) [55].

## 5. Conclusions

HY5 has a pivotal role in regulating anthocyanin accumulation in tomatoes. *Slhy5* mutants were created via the CRISPR/Cas9 system from ‘Indigo Rose’. *Slhy5* mutants displayed significantly lower anthocyanin accumulation than InR, both in seedlings and fruit. Interestingly, detectable levels of anthocyanins were present in hy5 mutant seedlings and fruit, and the pigment accumulation displayed obvious spatiotemporal specificity in the *Slhy5* seedling. These results indicated that other regulators existed to regulate anthocyanin biosynthesis in an HY5-independent manner. The total amount of flavonoids in InR was significantly higher than in the *Slhy5* mutant, and most of the genes associated with anthocyanin biosynthesis displayed higher expression levels in InR. *SlBBX24* likely regulated anthocyanin biosynthesis by interacting with *SlAN2-like*, *SlAN2*, while *SlWRKY44* interacted with *SlAN11*. Moreover, *SlPIF1* and *SlPIF3* seemed to be involved in anthocyanin biosynthesis by interacting with *SlBBX24*. We identified *SlBBX24* as a target to silence to produce fewer anthocyanins in tomato fruit peel, indicating the important role of *SlBBX24* in the regulation of anthocyanin accumulation. These results deepened the understanding of purple color formation in tomato seedlings and fruits in an *HY5*-dependent or independent manner via excavating the genes involved in anthocyanin biosynthesis. This study will help with the functional analysis of candidate genes controlling the color components of tomatoes and provide a theoretical basis for the breeding of tomatoes with high anthocyanin accumulation.

## Figures and Tables

**Figure 1 ijms-24-08690-f001:**
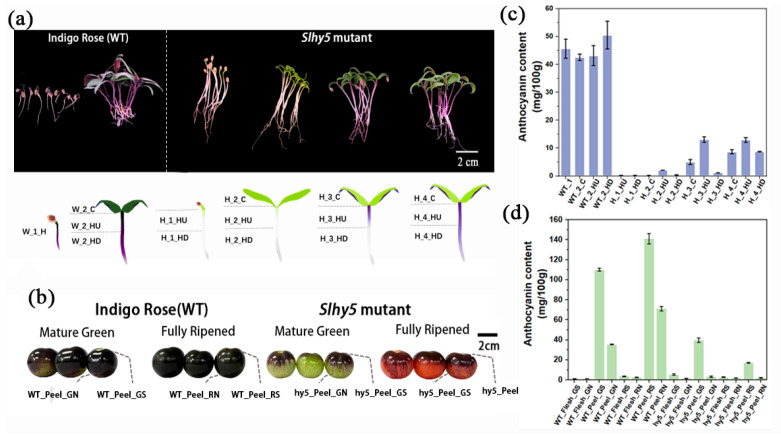
Morphological characterization of InR and *Slhy5* seedling and fruit. (**a**) Photograph showing the seedling phenotypes of different stages of InR and *Slhy5* seedlings for 4–7 days. (**b**) The phenotype of fruit from InR and *Slhy5* mutants at the mature green-mature stage and fully mature stage. (**c**) The anthocyanin content of InR and *Slhy5* seedling in different tissues. (**d**) The anthocyanin content of InR and *Slhy5* fruit s at the green-mature stage and fully ripened stage both in peel and flesh. Error bars indicate SD (*n* > 3).

**Figure 2 ijms-24-08690-f002:**
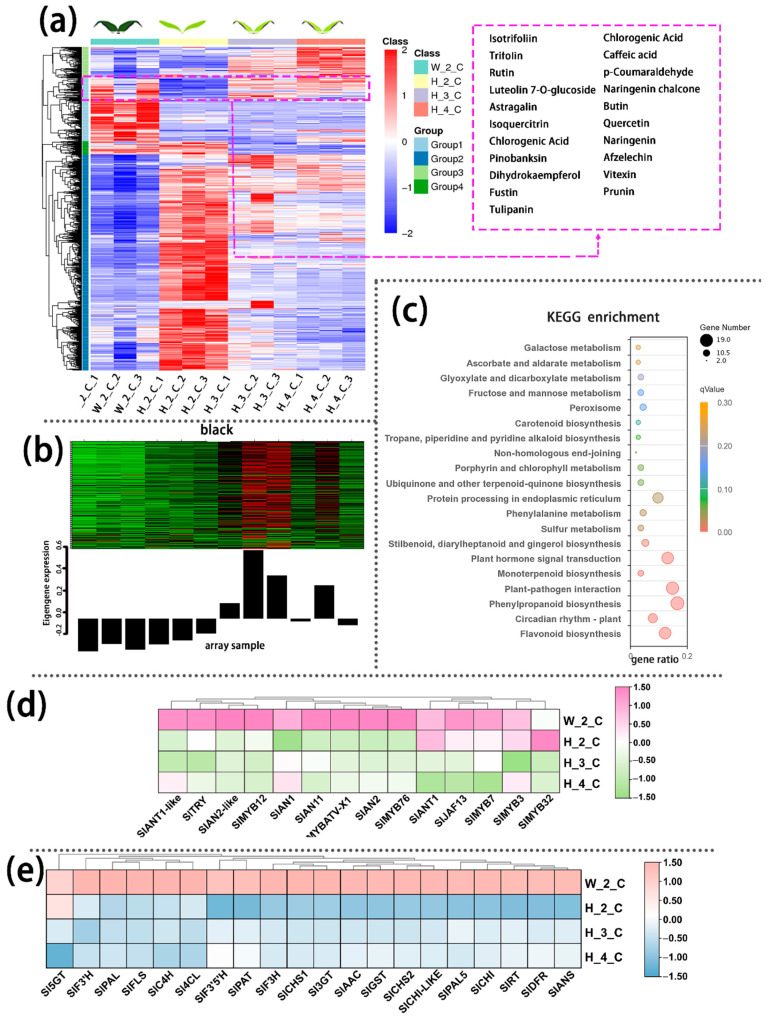
Differentially accumulated metabolites (DAMs) accumulation pattern consisted of the color variations during seedling development of cotyledon (**a**). The genes in the green module consisted of color variations during the seedling development of cotyledon (**b**). Top 20 enriched KEGG pathway enrichment of the genes in green module (**c**). The FKPM values of the transcriptional factors (**d**) and the structural genes (**e**) related to flavonoid and anthocyanin biosynthesis.

**Figure 3 ijms-24-08690-f003:**
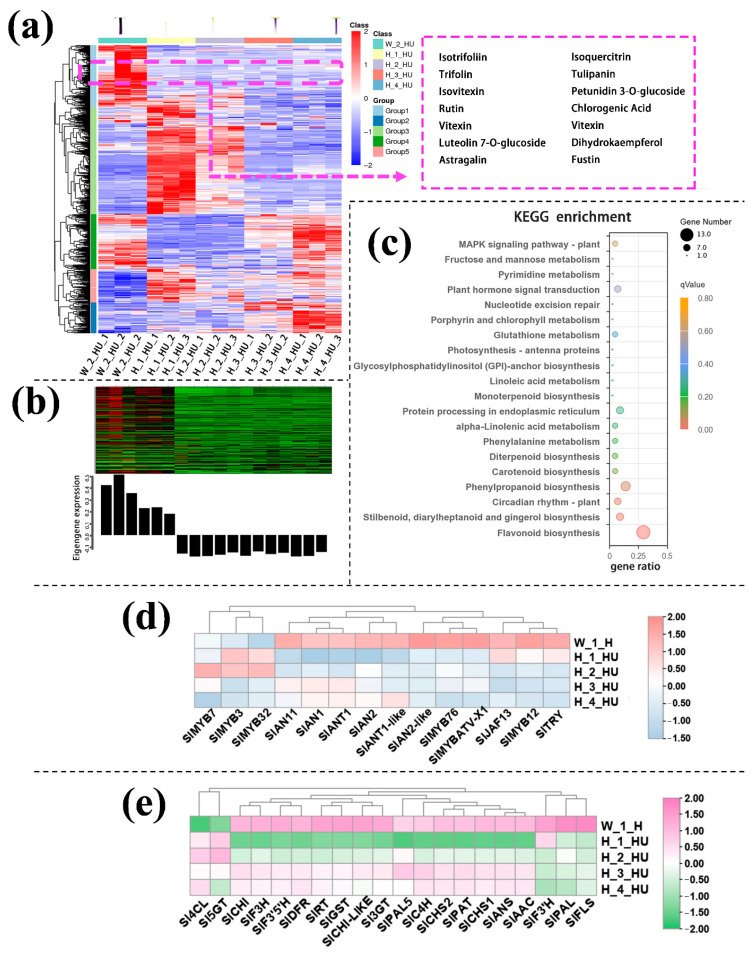
Differentially accumulated metabolites (DAMs) accumulation patterns consisted of color variations during seedling development of the upper part of the hypocotyl (**a**). The genes in the green module consisted of color variations during seedling development of the upper part of the hypocotyl (**b**). Top 20 enriched KEGG pathway enrichment of the genes in black module (**c**). The FKPM values of the transcriptional factors (**d**) and the structural genes (**e**) related to flavonoid and anthocyanin biosynthesis.

**Figure 4 ijms-24-08690-f004:**
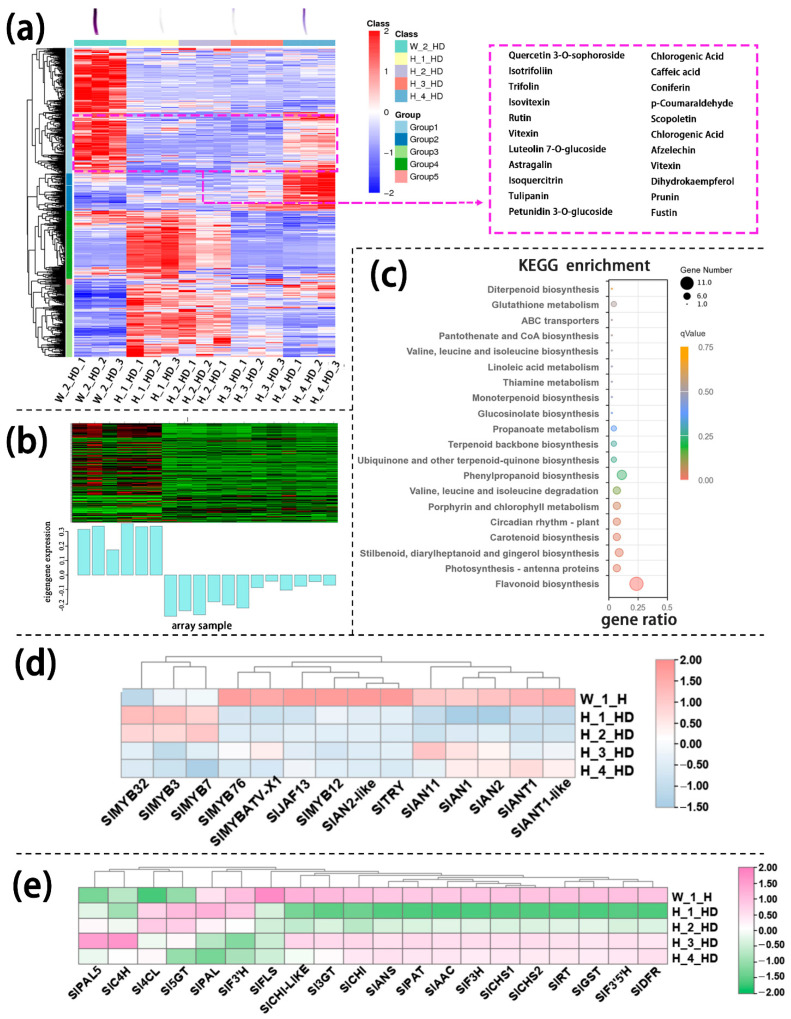
Differentially accumulated metabolites (DAMs) accumulation patterns consisted of color variations during seedling development of the lower part of the hypocotyl (**a**). The genes in the green module consisted of color variations during seedling development of the lower part of the hypocotyl (**b**). Top 20 enriched KEGG pathway enrichment of the genes in the pale turquoise module (**c**). The FKPM values of the transcriptional factors (**d**) and the structural genes (**e**) related to flavonoid and anthocyanin biosynthesis.

**Figure 5 ijms-24-08690-f005:**
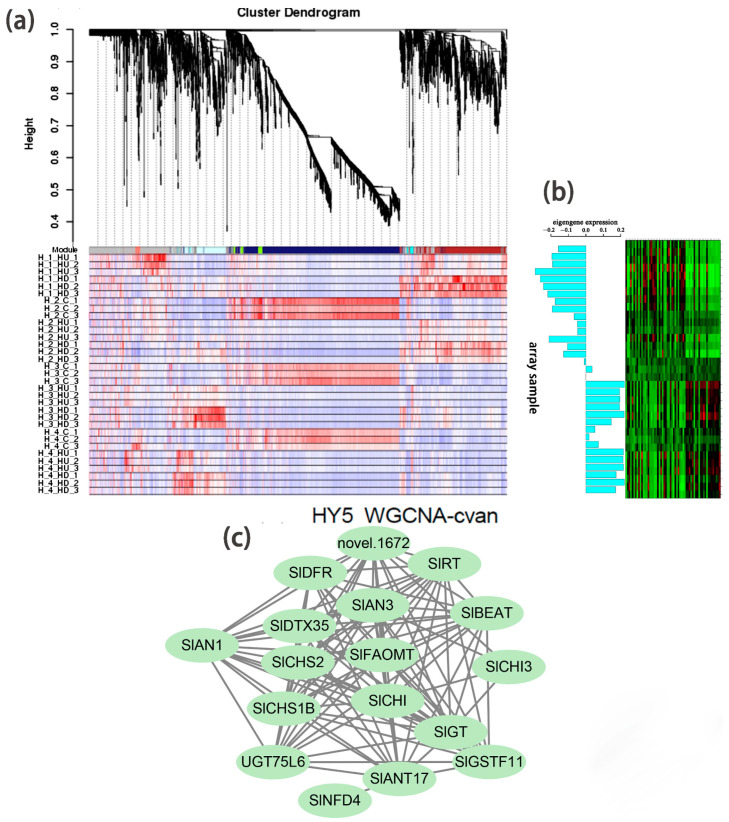
Hierarchical cluster tree showing seven modules obtained by WGCNA in *Slhy5* seedlings (**a**). Differential expression of genes in the accumulation pattern consisted of color variations during seedling development of *Slhy5* seedlings (**b**). Interaction network of DEG in the cyan model in *Slhy5* seedling (**c**).

**Figure 6 ijms-24-08690-f006:**
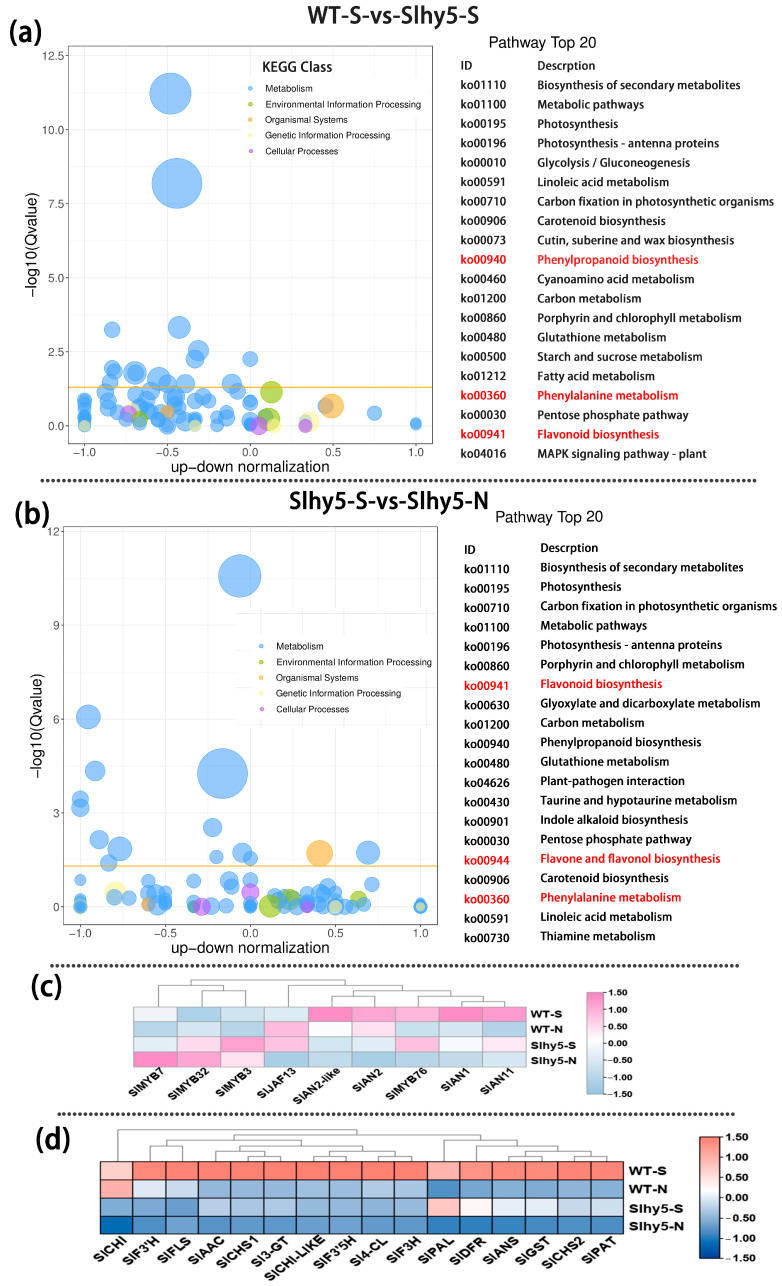
KEGG enrichment of differential genes in WT-S vs. Slhy5-S (**a**) and Slhy5-S vs. Slhy5-N (**b**). The FKPM values of the transcriptional factors (**c**) and the structural genes (**d**) related to flavonoid and anthocyanin biosynthesis.

**Figure 7 ijms-24-08690-f007:**
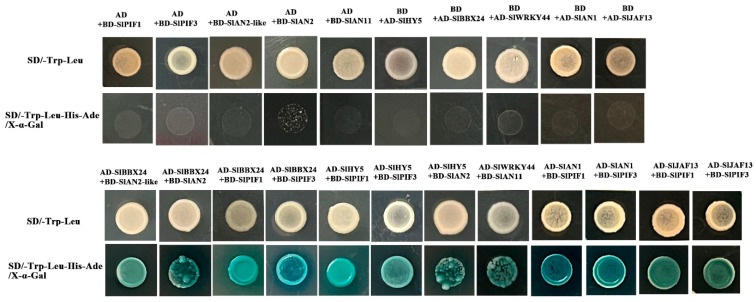
Yeast two-hybrid assay of the protein interactions between candidate genes involved in the components of the MBW complex that activates anthocyanidins in tomato.

**Figure 10 ijms-24-08690-f010:**
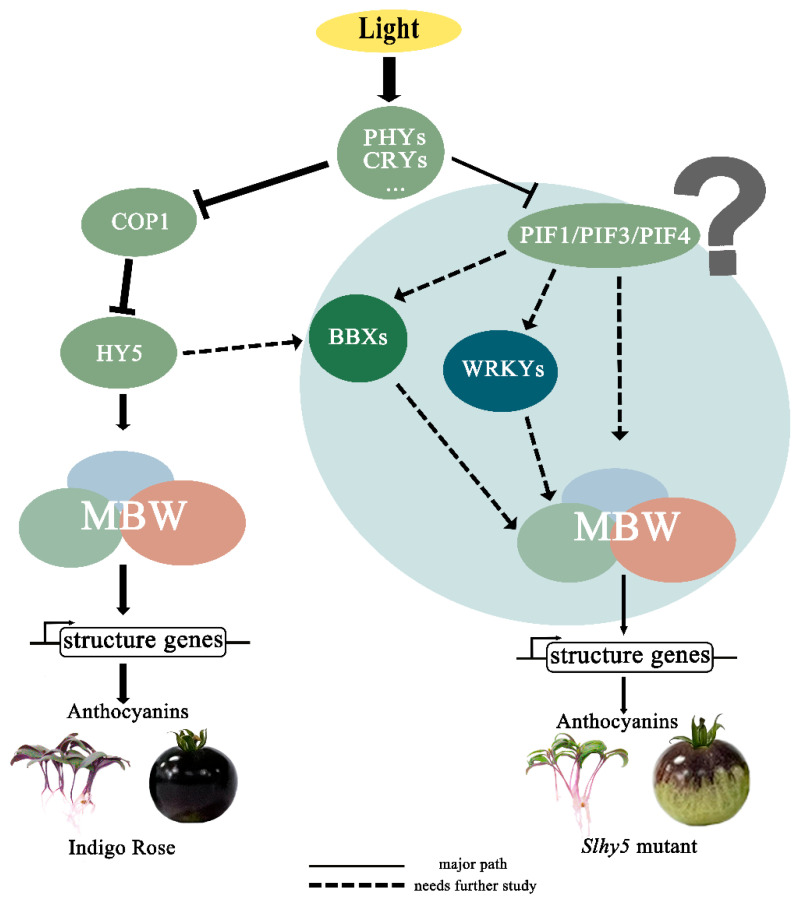
A putative model showing the anthocyanin induction pathways that might be dependent or independent of HY5 in tomatoes.

## Data Availability

Not applicable.

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
