# Peer review of "Omics Analysis Unveils the Pathway Involved in the Anthocyanin Biosynthesis in Tomato Seedling and Fruits"

_ijms, 2023, doi:10.3390/ijms24108690_

Round 1
Reviewer 1 Report
Dear Authors
In this study, it was performed omics analysis to clarify the regulatory network underlying anthocyanin biosynthesis in seedlings and fruit peels of ‘Indigo Rose’ and Slhy5 mutant. Results showed that the total amount of anthocyanins in both seedling and fruit of InR were significantly higher than those in Slhy5 mutant and most genes associated with anthocyanin biosynthesis exhibited higher expression levels in InR, suggesting that SlHY5 play pivotal roles in flavonoid bio-synthesis both in tomato seedlings and fruit.
The experimentation is interesting and was well conducted. The manuscript is well written, I do not suggest modifications in particular.
Minor editing of English language required.
Author Response
We appreciate your praise of our work and thank so much for your valuable and helpful comments
Reviewer 2 Report
Dear Authors and Editors,
The submitted manuscript examines the pathways of anthocyanin synthesis in specific variety of tomato in detail. A putative model showing the anthocyanin induction pathways was presented. Obtained results deepened the understanding of purple colour formation in ‘Indigo Rose’ variety tomato seedling and fruits in an HY5-dependent or independent manner via revealing the genes and mechanisms involved in anthocyanin biosynthesis based on omics analysis. Except that the Materials and Methods part is presented before the Results and not after the Discussion, the manuscript meets the requirements of the journal. Minor editorial corrections or clarifications are required:
Line 46 abbreviation MBW ?
Line 111 10/14 light/dark photoperiod? Maybe you can explain why this is used, and not, for example, 16/8?
FIG1A Should be W_2_HD in upper part of hypocotyl?
Line 176 words “the sequence” redundant
Line 200 Switzerland?
Section “Material and methods” according The Journal Int. J. Mol. Sci requirements should be placed after “Discussion”
Line 213 -217 revise sentence to make it better understandable
Line 237 Name of section: “3.2. Changes of metabolites and genes in the cotyledon of InR seedlings and Slhy5 seedlings” Better to change: “Changes of metabolites and gene expression in the cotyledon of InR seedlings and Slhy5 seedlings”
Line 311 in mode marked by paleturquoise colour?
313-314 The sentence “The KEGG enrichment analysis showed that the terms ‘flavonoid biosynthesis’ pathway was significantly enriched” can be changed as “The KEGG enrichment analysis showed that the ‘flavonoid biosynthesis’ pathway was significantly more pronounced” ?
Line 328-330 The sentence: “Based on WGCNA analysis, we identified a cyan module whose gene expression pattern was associated with the phenotype of anthocyanin synthesis in Slhy5 seedling at the third and four development stages (Figure 5a, b).” can be changed to: “Based on WGCNA analysis, we identified a module (marked in cyan), whose gene expression pattern was associated with the phenotype of anthocyanin synthesis in Slhy5 seedling at the third and four development stages (Figure 5a, b).”
Line 391 Revise title of section to make it better understandable
Line 406-407 “was silenced”
Line 501-504 Since according to the data provided by the literature and the authors, it is not the only regulator of tomato anthocyanin biosynthesis, maybe it is more correct to say that it is essential or one of most important in the regulatory chain, not the master regulator?
Since there is quite a lot of material provided, the authors simply need to calmly and carefully revise the text and correct minor linguistic inaccuracies.
Author Response
RESPONSES TO REVIEWERS COMMENTS
#Reviewer 2:
The submitted manuscript examines the pathways of anthocyanin synthesis in specific variety of tomato in detail. A putative model showing the anthocyanin induction pathways was presented. Obtained results deepened the understanding of purple colour formation in ‘Indigo Rose’ variety tomato seedling and fruits in an HY5-dependent or independent manner via revealing the genes and mechanisms involved in anthocyanin biosynthesis based on omics analysis. Except that the Materials and Methods part is presented before the Results and not after the Discussion, the manuscript meets the requirements of the journal. Minor editorial corrections or clarifications are required:
Response:We appreciate your praise of our work and thank so much for your valuable and helpful comments and suggestions to improve our paper. We have revised the manuscript accordingly. Our point-by-point responses are presented as follows.
Line 46 abbreviation MBW ?
Response:Thank you for your suggestion. We have added more details about MBW in the revised manuscript.(line 46)
Line 111 10/14 light/dark photoperiod? Maybe you can explain why this is used, and not, for example, 16/8?
Response:We truly appreciate your professional question. Based on our previous study result we found photoperiod of 10/14 light/dark is efficient enough for tomato seedling normal growth and photosynthesis.
FIG1A Should be W_2_HD in upper part of hypocotyl?
Response:We apologize for our carelessness. There should be W_2_HU and W_2_HD, which represent the upper part of hypocotyl and the lower part of hypocotyl respectively. We have corrected this mistake in the revised manuscript. (Figure 1.)
Line 176 words “the sequence” redundant
Response:Thank you for your professional suggestions. We have deleted the redundant word in the revised manuscript. (line672)
Line 200 Switzerland?
Response:We apologize for our carelessness. We have corrected this mistake in the revised manuscript. (line695 )
Section “Material and methods” according The Journal Int. J. Mol. Sci requirements should be placed after “Discussion”
Response:Thank you for pointing this out. The precedent version has been replaced.
Line 213 -217 revise sentence to make it better understandable
Response:We appreciate your valuable and helpful comments and suggestions to improve our paper. We have revised the manuscript accordingly. (line179-185)
Line 237 Name of section: “3.2. Changes of metabolites and genes in the cotyledon of InR seedlings and Slhy5 seedlings” Better to change: “Changes of metabolites and gene expression in the cotyledon of InR seedlings and Slhy5 seedlings”
Response:Thank you for pointing this out. We have modified the section in the revised manuscript according to your valuable suggestion. (line282)
Line 311 in mode marked by paleturquoise colour?
Response: We truly appreciate your professional question. The precedent version has been replaced. (line361)
313-314 The sentence “The KEGG enrichment analysis showed that the terms ‘flavonoid biosynthesis’ pathway was significantly enriched” can be changed as “The KEGG enrichment analysis showed that the ‘flavonoid biosynthesis’ pathway was significantly more pronounced” ?
Response:Thank you for your professional suggestions. We have revised the manuscript accordingly. (line365)
Line 328-330 The sentence: “Based on WGCNA analysis, we identified a cyan module whose gene expression pattern was associated with the phenotype of anthocyanin synthesis in Slhy5 seedling at the third and four development stages (Figure 5a, b).” can be changed to: “Based on WGCNA analysis, we identified a module (marked in cyan), whose gene expression pattern was associated with the phenotype of anthocyanin synthesis in Slhy5 seedling at the third and four development stages (Figure 5a, b).”
Response:Thank you for your professional suggestions. According to your suggestion, we have corrected this irrelevant part in the revised manuscript. (line381-384)
Line 391 Revise title of section to make it better understandable
Response:We appreciate your valuable and helpful comments and suggestions to improve our paper. We have revised the manuscript accordingly. (line452)
Line 406-407 “was silenced”
Response:Thank you for your professional suggestions. We have corrected and checked this mistake in the revised manuscript. (line471)
Line 501-504 Since according to the data provided by the literature and the authors, it is not the only regulator of tomato anthocyanin biosynthesis, maybe it is more correct to say that it is essential or one of most important in the regulatory chain, not the master regulator?
Response:We appreciate your praise of our work and thank so much for your valuable and helpful suggestions to improve our paper. We have revised the manuscript accordingly. (line479-480)
Reviewer 3 Report
Manuscript (ijms-2364246) “Omics Analysis Unveils the Pathway Involved in the Anthocyanin Bio-synthesis in Tomato Seedling and Fruits” by He et al. presents an interesting study about anthocyanin bio-synthesis and gene expression analysis in tomato.
This manuscript presents a valuable study with breeding applications. However, this manuscript is confused in some parts mainly in the description of the methodology and the experimental design. For these reasons, this manuscript is acceptable for publication in Antioxidants Journal after a major/moderate revision.
The major points for the revision of the manuscript are:
Around the whole manuscript authors must indicate the name of the genes in cursive.
Objectives are very large. Authors must simplify the objectives not including any methodological reference in the separate paragraph.
RNA-seq protocol should be explained indicating statistical analysis, bioinformatic analysis, reference genome, read length, biological and technical replications, etc.
qPCR protocol should be explained indicating statistical analysis, and biological and technical replications.
A new table or figure should be added with the qPCR validation of the RNA-Seq data. Main DEGs must be validated by using qPCR together with the two-hybrid assay.
Quality of Figures 2, 3, 4 and 5 must be improved increasing font size.
In the Conclusion section, authors must indicate the main implications of these results from an agronomical and breeding point of view.
English grammar and expression should be revised.
Author Response
RESPONSES TO REVIEWERS COMMENTS
#Reviewer 3:
The major points for the revision of the manuscript are:
Around the whole manuscript authors must indicate the name of the genes in cursive.
Response:Thank you for your professional suggestions. We have corrected and checked these mistakes in the revised manuscript.
Objectives are very large. Authors must simplify the objectives not including any methodological reference in the separate paragraph.
Response:Thank you for your professional suggestions. We have simplified the objectives in the revised manuscript. (line96-100)
RNA-seq protocol should be explained indicating statistical analysis, bioinformatic analysis, reference genome, read length, biological and technical replications, etc.
Response:Thank you for pointing this out. We have added more details about the RNA-seq protocol in the manuscript. Three independent biological replicates were performed. The RNA-seq sequencing and assembly of seedling and fruit peel were performed by NovoGene Science and Technology Corporation (Beijing, China) and Genedenovo Biotechnology Co., Ltd. (Guangzhou, China), respectively. The library preparations were sequenced on an Illumina Hiseq 4000 platform to generate paired-end reads. The raw sequence reads were filtered by removing adaptor sequences and low-quality sequence, and raw sequences were changed into clean reads. Then, the clean reads were then mapped to the tomato reference genome sequence (ITAG 4.0) (https://solgenomics.net/organism/Solanum_lycopersicum/genome/). (line627-644)
qPCR protocol should be explained indicating statistical analysis, and biological and technical replications.
Response:We apologize that the original submission was not clear for this important information. Three biological replicates and three qRT-PCR technical replicates were performed for each sample. The relative expression levels were normalized with the results of mean values of SlUBI using 2−ΔΔCt method. (line701-704)
A new table or figure should be added with the qPCR validation of the RNA-Seq data. Main DEGs must be validated by using qPCR together with the two-hybrid assay.
Response:Thank you for pointing this out. The figure of validation of RNA-Seq analysis via RT-qPCR has provided in the supplementary materials (Figure S8).
Quality of Figures 2, 3, 4 and 5 must be improved increasing font size.
Response:Thank you for your professional suggestions. We have improved our figure’s quality in the revised manuscript.
In the Conclusion section, authors must indicate the main implications of these results from an agronomical and breeding point of view.
Response:Thank you for your professional suggestions. We have indicated the main implications of these results from an agronomical and breeding point of view in the revised manuscript. (line725-730)
Round 2
Reviewer 3 Report
Authors have revised correctly the manuscript